# Multi-Intelligent Reflecting Surfaces and Artificial Noise-Assisted Cell-Free Massive MIMO Against Simultaneous Jamming and Eavesdropping

**DOI:** 10.3390/s24227326

**Published:** 2024-11-16

**Authors:** Huazhi Hu, Wei Xie, Kui Xu, Xiaochen Xia, Na Li, Huaiwu Wu

**Affiliations:** PLA Army Engineering University, Nanjing 210007, China; huhz121@163.com (H.H.); lgdxxukui@sina.com (K.X.); tjuxxc@sina.com (X.X.); linalala@sina.com (N.L.); zhangyue@gisinspection.com (H.W.)

**Keywords:** cell-free massive MIMO, intelligent reflecting surfaces, secure transmission, secrecy rate, artificial noise

## Abstract

In a cell-free massive multiple-input multiple-output (MIMO) system without cells, it is assumed that there are smart jammers and disrupters (SJDs) that attempt to interfere with and eavesdrop on the downlink communications of legitimate users. A secure transmission scheme based on multiple intelligent reflecting surfaces (IRSs) and artificial noise (AN) is proposed. First, an access point (AP) selection strategy based on user location information is designed, which aims to determine the set of APs serving users. Then, a joint optimization framework based on the block coordinate descent (BCD) algorithm is constructed, and a non-convex optimization solution based on the univariate function optimization and semi-definite relaxation (SDR) is proposed with the aim of maximising the minimum achievable secrecy rate for users. By solving the univariate function maximisation problem, the multi-variable coupled non-convex problem is transformed into a solvable convex problem, obtaining the optimal AP beamforming, AN matrix and IRS phase shift matrix. Specifically, in a single-user scenario, the scheme of multiple intelligent reflecting surfaces combined with artificial noise can improve the user’s achievable secrecy rate by about 11% compared to the existing method (single intelligent reflective surface combined with artificial noise) and about 2% compared to the scheme assisted by multiple intelligent reflecting surfaces without artificial noise assistance.

## 1. Introduction

The natural openness and broadcast nature of the wireless channel pose a significant security risk to wireless transmissions. In wireless communication systems, interference and eavesdropping can cause serious practical harm, such as reducing communication quality, causing communication disruptions, leaking personal privacy, stealing trade secrets, hindering industrial production, and threatening national security. In the 6G network, anti-interference has many potential application scenarios, such as being used in intelligent transportation systems, the industrial Internet of Things, safeguarding military operations, protecting the core secrets of enterprises, the financial industry, and the government, and enhancing personal privacy communications. The jammer sends jamming information to the user by means of full duplex techniques and by adjusting its precoding [1] in order to reduce the signal-to-noise ratio (SINR) of the destination user. Active or passive Eve uses the propagation characteristics of the wireless channel to steal information from legitimate users, resulting in system information leakage. Traditional approaches focus on enhancing the physical layer security of the system by intelligently designing transmission coding strategies, developing keys on the common channel and establishing collaborative contract relays [2,3,4]. The literature [5] proposed to use the random fading property of the wireless channel to protect the information transmission and prevent the information access by illegal receivers. Based on Shannon’s information theory, Wyner was the first to propose the concept of secrecy capacity in [6] as a measure of system security performance. In [7], the authors proved that the secrecy capacity of the system in the Gaussian noise channel model is equal to the difference between the legitimate user channel and the eavesdropping channel. The authors in [8] extended this conclusion to broadcast channels and Rayleigh fading channels. Since then, the wire-tap model has been widely used to study secure communications in eavesdropping and jamming scenarios, mainly involving multiple-input single-output (MISO), multiple-input multiple-output (MIMO), millimeter-wave and terahertz scenarios.

Massive MIMO, a key technology for 5G mobile communications, has been a hot topic of research in academia. Massive MIMO uses the characteristics of “channel solidification” and “favorable propagation” to reduce the complexity of signal processing and significantly improve the spectral efficiency and energy efficiency of communication systems [9,10]. Massive MIMO has also been of concern in terms of physical layer security. On the one hand, an attacker can take advantage of the high array gain of massive MIMO, which is sensitive to channel estimation error, to send malicious pilot sequences to increase the channel estimation error and induce the base station beamforming to point at the attacker, thus reducing the security capacity of the system [11]. On the other hand, it is possible for attackers to observe the spectrum data of legitimate users by monitoring base stations continuously. They can then use machine learning, full-duplex technology and other techniques to disguise themselves more convincingly and employ intelligent interference methods, which makes secure transmission in large-scale MIMO scenarios even more challenging. With the rapid development of artificial intelligence, smart jamming techniques have been further studied. The existing intelligent interference technology is a multiple overlapping of cognitive interference and intelligent interference in terms of connotation and extension [12]. Most scholars study only from the theory of cognitive radio and jamming strategy, and there are fewer studies on smart jamming devices (intelligent devices integrating jamming and eavesdropping). Intelligent jamming devices can masquerade as legitimate users to steal information transmitted from the base station to legitimate users, and it can also send malicious messages to interfere with the reception of legitimate users. Current anti-jamming techniques only consider scenarios of jamming or eavesdropping, and rarely study secure transmission in harsh communication environments where both jamming and eavesdropping are present.

Cell-free massive MIMO is based on the idea of breaking the “cell-centric” architecture and adopting a “user-centric” network architecture, effectively avoiding the “edge effect” of cellular cells and the signaling interactions caused by cell switching [13,14]. The differentiated network architecture and incremental signal processing methods introduce novel security concerns at the physical layer of de-cellular massive MIMO.

### 1.1. Motivations

At present, research on cell-free massive MIMO secure transmission is still in its infancy. Ref. [15] addressed the secure transmission problem in cellular massive MIMO under Rayleigh channel conditions with multiple eavesdroppers. The authors derive a closed-form expression for the achievable secrecy rate based on an additive white Gaussian noise model. Ref. [16] studied secure transmission under Rician channel conditions in the presence of multiple antenna Eve in cell-free massive MIMO, relying on an additive quantized noise model to derive a closed-form expression for the achievable secrecy rate. Ref. [17] proposed a full-duplex based cell-free massive MIMO anti-eavesdropping method to maximize the system secrecy capacity by optimizing the duplex selection mode and secrecy transmission. Ref. [18] studied the downlink confidential transmission of the system under energy recovery constraints and proposed a transmission power optimization scheme based on semi-definite relaxation planning. Refs. [19,20,21] investigated the problem of secrecy optimization for resisting active eavesdropping attacks under the conditions of spatially correlated Riley fading, transceiver hardware damage, and identification of guided frequency spoofing attacks, respectively.

An Intelligent Reflecting Surface (IRS) is considered to be a key technology for 6G, and research into its relevance to wireless communications is in full swing. Ref. [22] investigated IRS-assisted single-input single-output (SISO) communication with multiple single-antenna users to minimize transmitted power by jointly optimizing the base station precoding matrix and the IRS reflected phase. Ref. [23] proposed to design the base station transmit power and IRS phase shift to improve the energy efficiency of the system while ensuring the direct link boundary between the user and the base station.The literature [24] proposes a signal beam routing methodology utilizing multiple IRS reflections with the objective of investigating wireless power transfer from a multiple-antenna base station to multiple energy users (EUs). The proposal set forth in the literature [25] is to provide support for disparate user groups through the deployment of multiple RSs. This approach effectively mitigates interference and leverages secondary reflections between IRSs, enhancing the data rate for each user. Research on IRS-assisted communications has become progressively more extensive and in-depth, focusing on IRS-assisted channel estimation [26,27], beamforming and reflection phase shift optimization [22,28,29], and deployment strategies [30,31].

The enhancement of wireless communication physical layer security by the IRS represents a promising avenue of research, although it is still in its nascent stages. Ref. [32] investigated single-antenna Eve scenarios in which IRS-assisted multi-antenna AP send confidential information to users. AP beamforming and IRS phase-shift are jointly designed to maximize the secrecy rate under the constraints that the channel space is highly correlated and the eavesdropping channel is stronger than the legitimate channel. Ref. [33] investigated the channel state information (CSI) of imperfectly known multi-antenna Eve and jointly optimized artificial noise (AN), beamforming and IRS reflection phase shift sent by base stations to build physical layer security in a radio environment. Refs. [34,35] reviewed the latest applications of IRS in future wireless networks and the design of secure transmissions, using IRS to modulate uncontrollable radio fading channels and build intelligent radio environments. Current research scenarios on IRS-assisted secure communication are dominated by MISO, MIMO, non-orthogonal multiple access (NOMA), and millimeter wave. Research combining IRS with cell-free massive MIMO has also only appeared in [36,37,38] but is also limited to assisted augmented communication, and exploration regarding secure transmission has yet to occur.

### 1.2. Contributions

For these reasons, this paper initially explores the potential of multi-IRS assisted cell-free massive MIMO secure transmission. The multi-IRS localization method of [39] is used to initially determine the legitimate user location information, and using a large-scale fading-based AP selection strategy, the user’s service AP set is determined. The AP emission precoding, AN matrix and IRS reflection phase shift are designed collaboratively to maximize the minimum secrecy rate of the user. The main work is as follows.

We construct a joint framework for multi-IRS and AN-assisted cell-free massive MIMO downlink robust and secure communication, proposing a model to resist transient interference and eavesdropping.Determining the initial location information of legitimate users through IRS localization methods and determining the service AP set of users based on a large-scale fading AP selection strategy.The joint design of AP beamforming, AN matrix and IRS reflection phase shift to maximize the minimum achievable secrecy rate of the user under the constraint of satisfying the total power and IRS unit modulus.The numerical simulation experimental results demonstrate the effectiveness of the proposed multi-IRS joint AN scheme, which provides ideas for single-IRS to multi-IRS assisted physical layer security research.

### 1.3. Methods

In order to reduce the interference to downlink users, we propose to jointly design an intelligent reflecting phase and artificial noise matrix. A joint optimization framework based on the block coordinate descent algorithm is designed with the aim of maximizing the minimum achievable secrecy rate of the users, and a non-convex optimization solution method based on univariate function optimization and semi-definite relaxation is formulated. Specifically, based on the BCD algorithm, the optimal AP beamforming, AN matrix and IRS phase shift matrix are obtained by solving the single-variable function-maximum problem and transforming the multivariable coupled non-convex problem into a solvable convex problem.

### 1.4. Organizations

The main structure of this paper is as follows. Section 2 shows the system model for multi-IRS assisted cell-free massive MIMO secure transmission. In Section 3, we describe the problem and propose an objective function for optimization. In Section 4, we present a joint precoding and AN design framework for maximizing the minimum achievable secrecy rate for the user. The results of simulation experiments and discussion are presented in Section 5 to verify the performance of the proposed scheme. Finally, conclusions are drawn in Section 6.

### 1.5. Notations

In the paper, lowercase bold letters represent a vector, while uppercase bold letters represent a matrix. ${\left[\cdot \right]}^{-1}$, ${\left[\cdot \right]}^{T}$, and ${\left[\cdot \right]}^{H}$ denote the inverse, transpose, and conjugate transpose operations, respectively. $C^{M\times N}$ represents the set of all M×N complex-valued matrices. |*a*| denotes the Euclidean norm of *a*. **A** ⪰ 0 indicates that **A** is a positive semidefinite matrix. *Tr*(**A**) represents the trace of matrix **A**. E[·] is the expectation operator, and *diag*(·) represents the diagonal operation **0***_L_* denotes a column vector of length *L* with all elements equal to 0. 

## 2. System Model

### 2.1. Channel Model

As shown in Figure 1, multiple IRSs assisted cell-free massive MIMO secure communication networks. *L* multi-antenna APs jointly serve *K* single-antenna legal users through *I* IRSs. There is a smart jamming device (SJD) integrating jamming and eavesdropping in the system, trying to interfere and eavesdrop on the information of legitimate users. Both APs and SJD can transmit information to users through two paths, direct channel or cascaded channel. Each AP is equipped with *Q* antennas, and the SJD is equipped with $Q_{e}+1$ antennas. The $Q_{e}$ antennas is used to jam users, and the other antenna is disguised as a single-antenna user to steal legitimate user information. Consider deploying *I* IRS, each with *N* antennas. The AP and IRS are connected to the central processing unit (CPU) through the fronthaul link, and the CPU jointly designs the AP beamforming, AN matrix and IRS phase-shift matrix to resist malicious interference and eavesdropping of SJD.

As shown in Figure 2, $h_{k,l}^{H}\in C^{1\times Q}$ represents the direct channel between the *l*-th AP and the *k*-th user, $g_{k,i}^{H}\in C^{1\times N}$ represents the channel between the *i*-th IRS and the *k*-th user, and $G_{i,q}\in C^{N\times Q}$ represents the channel between the *q*-th AP and the *i*-th IRS. For the SJD, $h_{Eve,l}^{H}\in C^{1\times Q}$ represents the channel between the *q*-th AP and the Eve, $g_{Eve,i}^{H}\in C^{1\times N}$ represents the channel between the *i*-th IRS and the Eve, $G_{i,Ja}\in C^{N\times Q_{e}}$ represents the channel between the jammer and the *i*-th IRS, and $h_{k,Ja}^{H}\in C^{1\times Q_{e}}$ represents the channel between the jammer and the *k*-th user. Assuming that the integrated Eve is not affected by the information sent by the jammer, the interference of the jammer to the Eve can be counteracted by adopting self-interference cancellation technology [32].

### 2.2. Maximum Large-Scale Fading-Based AP Selection Strategy

Cell-free massive MIMO breaks the network architecture of the cellular and provides uniform signal coverage for users in the area with massive APs, and cooperative signal processing is completed by the CPU. Because the coverage area is wide and APs are not evenly distributed, as the number of users increases, the centralized signal processing method will bring huge backhaul link load, and the computational complexity of the CPU beamforming design will greatly increase [40]. Relying on the multi-IRS joint positioning method, it assumes that the rough location information of legitimate users is known [39]. Using the large-scale fading information between APs and users, an effective AP selection strategy is constructed for each user, and its service set is determined. The CPU collaboratively designs the beamforming according to the users served by each AP.

Let $\left|M_{k}\right|=L_{k}<L$ represent the number of APs serving the *k*-th user, where $M_{k}\subset \left\{1,2,\ldots ,L\right\}$ represents the service set of the *k*-th user, which satisfies the following conditions:(1)$$\sum_{l=1}^{L_{k}}\left(\frac{{\bar{\beta }}_{lk}}{\sum_{m=1}^{L}\beta_{mk}}\right)⩾\partial \%$$
where $\left\{\beta_{1k},\ldots ,\beta_{Lk}\right\}$ represents the large-scale fading of each *k*-th user, *L* APs, $\left\{{\bar{\beta }}_{1k},\ldots ,{\bar{\beta }}_{Lk}\right\}$ represents the set in descending order, and *∂* represents the threshold limit. The service AP set of each user can be filtered by the above strategy.

### 2.3. Downlink Transmission

It can be seen from the above that the information received by the *k*-th legitimate user and the Eve can be written as
(2)$$\begin{array}{c} \begin{array}{cc} y_{k} & =\left(\sum_{l\in M_{k}}\left(\sum_{i=1}^{I}g_{k,i}^{H}\Theta_{i}G_{i,l}+h_{k,l}^{H}\right)\left(w_{k,l}d_{k,l}+z_{k,l}\right)\right)\\ \; & +\sum_{j\neq k}^{K}\left\{\sum_{l\in M_{j}}\left(\sum_{i=1}^{I}g_{k,i}^{H}\Theta_{i}G_{i,l}+h_{k,l}^{H}\right)\left(w_{j,l}d_{j,l}+z_{j,l}\right)\right\}\\ \; & +\sum_{j=1}^{K}\left(\sum_{i=1}^{I}g_{k,i}^{H}\Theta_{i}G_{i,Ja}+h_{k,Ja}^{H}\right)w_{j,Ja}d_{j,Ja}+n_{k} \end{array} \end{array}$$
(3)$$\begin{array}{c} \begin{array}{cc} y_{Eve}^{k} & =\left(\sum_{l\in M_{k}}\left(\sum_{i=1}^{I}g_{Eve,i}^{H}\Theta_{i}G_{i,l}+h_{Eve,l}^{H}\right)\left(w_{k,l}d_{k,l}+z_{k,l}\right)\right)\\ \; & +\sum_{j\neq k}^{K}\left\{\sum_{l\in M_{j}}\left(\sum_{i=1}^{I}g_{Eve,i}^{H}\Theta_{i}G_{i,l}+h_{Eve,l}^{H}\right)\left(w_{j,l}d_{j,l}+z_{j,l}\right)\right\}\\ \; & +n_{Eve} \end{array} \end{array}$$
where $\Theta_{i}$ represents the reflection phase matrix of the *i*-th IRS, and $\Theta_{i}=\sqrt{\eta }diag$$\left({\left[e^{j\theta_{i,1}},e^{j\theta_{i,2}},\ldots e^{j\theta_{i,N}}\right]}^{H}\right)$, η is the IRS reflection coefficient. In practice, the effect of the IRS reflection amplitude on the system has been studied [41]. In this paper, only the influence of the IRS phase-shift matrix is considered. For the convenience of illustration, we set the reflection amplitude to 1. $\theta_{i,n}$ is the phase shift of the *n*-th reflective element of the *i*-th IRS. $w_{k,l}\in C^{Q\times 1}$ represents the precoding matrix of the *l*-th AP to the *k*-th user, $z_{k,l}\in C^{Q\times 1}$ represents the AN matrix of the *l*-th AP to the *k*-th user, $w_{k,Ja}\in C^{Q_{e}\times 1}$ represents the precoding matrix of the jammer to the *k*-th user. $d_{j,l}$ and $d_{j,Ja}$ represent the information transmitted by the *l*-th AP and jammer to user j, respectively. $\Xi \left[{\left|d_{j,l}\right|}^{2}\right]=\Xi \left[{\left|d_{j,Ja}\right|}^{2}\right]=1$. $n_{k},n_{Eve}∼CN\left(0,N_{0}\right)$ represent the Gaussian additive white noise at the receiver of user *k* and the Eve, respectively, with the mean of 0 and the variance of $N_{0}$. Considering that the jammer does not need to manage energy leakage and inter-user interference, the classical matched filter architecture is adopted, and its precoding matrix can be expressed as
(4)$$\begin{array}{c} \begin{array}{c} w_{kJa}=\sqrt{p_{Ja,k}}\frac{{\hat{H}}_{k,Ja}^{H}}{∥{\hat{H}}_{k,Ja}^{H}∥},\;k\in \left\{1,\ldots ,K\right\} \end{array} \end{array}$$
where $H_{k,Ja}^{H}=\sum_{i=1}^{I}g_{k,i}^{H}\Theta_{i}G_{i,Ja}+h_{k,Ja}^{H}$ is composed of a jammer-IRS-user cascade channel and direct channel, and ${\hat{H}}_{k,Ja}^{H}$ represents the channel estimation between the jammer and user.

## 3. Problem Formulation

Based on the AP selection strategy, in this section, we consider the joint design of AP beamforming, the AN matrix, and the IRS phase shift matrix under the constraints of AP transmit power, using an IRS unit modulus to maximize the minimum achievable secrecy rate. For convenience, we define $f_{k,l}^{H}=\sum_{i=1}^{I}g_{k,i}^{H}\Theta_{i}G_{i,l}+h_{k,l}^{H}\;$, $f_{Eve,l}^{H}=\sum_{i=1}^{I}g_{Eve,i}^{H}\Theta_{i}G_{i,l}+h_{Eve,l}^{H}$, $f_{k,l}^{H},f_{Eve,l}^{H}\in C^{1\times Q}$, $l\in M_{k}$. Equations (2) and (3) can be rewritten as
(5)$$\begin{array}{c} \begin{array}{cc} y_{k} & =\sum_{l\in M_{k}}f_{k,l}^{H}\left(w_{k,l}d_{k,l}+z_{k,l}\right)\\ \; & +\sum_{j\neq k}^{K}\left(\sum_{l\in M_{j}}f_{k,l}^{H}\left(w_{j,l}d_{j,l}+z_{j,l}\right)\right)+\sum_{j=1}^{K}H_{k,Ja}^{H}x_{k,Ja}+n_{k} \end{array} \end{array}$$
(6)$$\begin{array}{c} \begin{array}{cc} y_{Eve}^{k} & =\sum_{l\in M_{k}}f_{Eve,l}^{H}\left(w_{k,l}d_{k,l}+z_{k,l}\right)\\ \; & +\sum_{j\neq k}^{K}\left(\sum_{l\in M_{j}}f_{Eve,l}^{H}\left(w_{j,l}d_{j,l}+z_{j,l}\right)\right)+n_{Eve} \end{array} \end{array}$$

From the above, the SINR of user *k* and Eve can be written as
(7)$$\begin{array}{c} \begin{array}{c} SINR_{k}=\frac{{\left|\sum_{l\in M_{k}}f_{k,l}^{H}w_{k,l}\right|}^{2}}{A} \end{array} \end{array}$$
where
(7a)$$\begin{array}{c} \begin{array}{cc} A & =\sum_{j\neq k}^{K}{\left|\sum_{l\in M_{j}}f_{k,l}^{H}w_{j,l}\right|}^{2}+\sum_{j=1}^{K}{\left|\sum_{l\in M_{j}}f_{k,l}^{H}z_{j,l}\right|}^{2}\\ \; & +{\left|\sum_{j=1}^{K}H_{k,Ja}^{H}w_{j,Ja}\right|}^{2}+N_{0} \end{array} \end{array}$$
(8)$$\begin{array}{c} \begin{array}{cc} \; & SINR_{Eve}^{k}={\left|\sum_{l\in M_{k}}f_{Eve,l}^{H}w_{k,l}\right|}^{2}\\ \; & \times {\left\{\sum_{j\neq k}^{K}{\left|\sum_{l\in M_{j}}f_{Eve,l}^{H}w_{j,l}\right|}^{2}+\sum_{j=1}^{K}{\left|\sum_{l\in M_{j}}f_{Eve,l}^{H}z_{j,l}\right|}^{2}+N_{0}\right\}}^{-1} \end{array} \end{array}$$
Defining $f_{k}^{H}=\left[f_{k,1}^{H},\ldots ,f_{k,L}^{H}\right]$, $f_{Eve}^{H}=\left[f_{Eve,1}^{H},\ldots ,f_{Eve,L}^{H}\right]$, ${\tilde{w}}_{k}={\left[w_{k,1},\ldots w_{k,l},w_{k,L}\right]}^{H}$, ${\tilde{z}}_{k}={\left[z_{k,1},\ldots z_{k,l},z_{k,L}\right]}^{H}$, $\tilde{\Theta }=\left[\Theta_{1},\Theta_{2},\ldots ,\Theta_{I}\right]$, where
(9)$$\begin{array}{c} \begin{array}{c} w_{k,l}=\left\{\begin{array}{c} w_{k,l}\;,\;l\in M_{k}\\ 0_{Q}\;,\;l\notin M_{k} \end{array}\right. \end{array} \end{array}$$
(10)$$\begin{array}{c} \begin{array}{c} z_{k,l}=\left\{\begin{array}{c} z_{k,l}\;,\;l\in M_{k}\\ 0_{Q}\;,\;l\notin M_{k} \end{array}\right. \end{array} \end{array}$$

Equations (9) and (10) characterize that each user is served by the selected AP set. When AP *l* does not belong to the serving set of user k, its corresponding precoding and artificial noise are $0_{Q}$, and $0_{Q}$ represents a 0 vector with dimension Q×1.

Therefore, the SINR of user *k* and Eve can be further formulated as
(11)$$\begin{array}{c} SINR_{k}=\frac{Tr\left(F_{k}W_{k}\right)}{\sum_{j\neq k}^{K}Tr\left(F_{k}W_{j}\right)+\sum_{k=1}^{K}Tr\left(F_{k}Z_{k}\right)+J_{k}} \end{array}$$
(12)$$\begin{array}{c} SINR_{Eve}^{k}=\frac{Tr\left(F_{Eve}W_{k}\right)}{\sum_{j\neq k}^{K}Tr\left(F_{Eve}W_{j}\right)+\sum_{k=1}^{K}Tr\left(F_{Eve}Z_{k}\right)+N_{0}} \end{array}$$
where $F_{k}=f_{k}f_{k}^{H}$, $F_{Eve}=f_{Eve}f_{Eve}^{H}$, $W_{k}={\tilde{w}}_{k}{\tilde{w}}_{k}^{H}$, $Z_{k}={\tilde{z}}_{k}{\tilde{z}}_{k}^{H}$, $J_{k}={\left|\sum_{j=1}^{K}H_{k,Ja}^{H}w_{j,Ja}\right|}^{2}+N_{0}$. The downlink achievable rate of user *k* and Eve are $R_{k}^{d}=\log_{2}\left(1+SINR_{k}\right)$ and $R_{E\mathrm{ve},k}^{d}=\log_{2}\left(1+SINR_{E\mathrm{ve}}^{k}\right)$, respectively. For user *k*, its security capacity is
(13)$$\begin{array}{c} R_{\sec,k}^{d}={\left[R_{k}^{d}-R_{E\mathrm{ve},k}^{d}\right]}^{+}={\left[\log_{2}\left(\frac{1+SINR_{k}}{1+SINR_{E\mathrm{ve}}^{k}}\right)\right]}^{+} \end{array}$$

In order to ensure the security requirements of each legitimate user in the system, it is necessary to prevent information leakage caused by the interference of SJD on a user beyond expectations. We aim to improve the worst performance of legitimate users in the system, that is, to maximize the minimum achievable secrecy rate for legitimate users. Jointly optimizing AP beamforming, an AN matrix, and an IRS phase shift matrix, the optimization problem can be expressed as
(14)$$\begin{array}{ll} P1: & \max_{W,\tilde{\Theta },Z}\;\min_{k}\;R_{\sec,k}^{d}\\ s.t. & C1:Tr\left(W_{k}\right)+Tr\left(Z_{k}\right)⩽P_{\max}\\ & W_{k}⪰0,\;Z_{k}⪰0,\;k\in \left\{1,2,\ldots ,K\right\}\\ & C2:0⩽\theta_{n}⩽2\pi ,\;\left|e^{j\theta_{n}}\right|\;=\;1,\;n\in \left\{1,\ldots ,N\right\} \end{array}$$
where C1 denotes the transmit power constraint, and C2 represents the element unit modulus constraint of the IRS reflection.

## 4. Joint Precoding and an Design

Generally speaking, the maximum and minimum optimization problem is a complexity solving process. P1 is the optimization problem of multi-variable coupling of W, Z, $\tilde{\Theta }$, which further increases the difficulty of solving the problem. Inspired by the alternate optimization method, when the other two variables are fixed, P1 is transformed into an optimization problem related to only one variable, and the complexity of the solution is greatly reduced.

In this subsection, we adopt the block coordinate descent (BCD) algorithm to transform the original complex coupled puzzle into two solvable suboptimal solutions. When solving the optimal solution of each variable, in order to avoid the high complexity brought by the successive convex approximation (SCA) algorithm, we use an ingenious function transformation method to convert the complex problem of subtracting two logarithms into a single function maximum value problem, relax the non-convex constraint to transform it into a solvable convex problem, and finally obtain the respective optimal solutions.

### 4.1. Sub-Problem for W, Z When Given $\tilde{\Theta }$

Constraint C2 can be removed for a given IRS phase shift. For further illustration, we expand Equation (13) at the top of this page. Here, $\varepsilon_{0}=\frac{1}{J_{k}}$, $\varepsilon_{1}=\frac{1}{N_{0}}$. $\chi_{k}^{1}$, $\chi_{Eve,k}^{2}$, $\chi_{k}^{3}$ and $\chi_{Eve,k}^{4}$ are represented as follows
(15)$$\begin{array}{c} \begin{array}{cc} R_{k}^{d}-R_{Eve}^{d} & =\log_{2}\left(\frac{\varepsilon_{0}\sum_{j=1}^{K}\mathrm{Tr}(F_{k}W_{j})+\gamma_{0}\sum_{j=1}^{K}\mathrm{Tr}(F_{k}Z_{j})+1}{\varepsilon_{0}\sum_{j=1,j\neq k}^{K}\mathrm{Tr}(F_{k}W_{j})+\gamma_{0}\sum_{j=1}^{K}\mathrm{Tr}(F_{k}Z_{j})+1}\times \frac{\varepsilon_{1}\sum_{j=1,j\neq k}^{K}\mathrm{Tr}(F_{Eve}W_{j})+\gamma_{1}\sum_{j=1}^{K}\mathrm{Tr}(F_{Eve}Z_{j})+1}{\varepsilon_{1}\sum_{j=1}^{K}\mathrm{Tr}(F_{Eve}W_{j})+\gamma_{1}\sum_{j=1}^{K}\mathrm{Tr}(F_{Eve}Z_{j})+1}\right)\\ \; & =\log_{2}\left(\chi_{k}^{1}\right)+\log_{2}\left(\chi_{Eve,k}^{2}\right)-\log_{2}\left(\chi_{k}^{3}\right)-\log_{2}\left(\chi_{Eve,k}^{4}\right) \end{array} \end{array}$$
(16)$$\begin{array}{c} \chi_{k}^{1}=\varepsilon_{0}\sum_{j=1}^{K}\mathrm{Tr}(F_{k}W_{j})+\gamma_{0}\sum_{j=1}^{K}\mathrm{Tr}(F_{k}Z_{j})+1 \end{array}$$
(17)$$\begin{array}{c} \chi_{Eve,k}^{2}=\varepsilon_{1}\sum_{j=1,j\neq k}^{K}\mathrm{Tr}(F_{Eve}W_{j})+\gamma_{1}\sum_{j=1}^{K}\mathrm{Tr}(F_{Eve}Z_{j})+1 \end{array}$$
(18)$$\begin{array}{c} \chi_{k}^{3}=\varepsilon_{0}\sum_{j=1,j\neq k}^{K}\mathrm{Tr}(F_{k}W_{j})+\gamma_{0}\sum_{j=1}^{K}\mathrm{Tr}(F_{k}Z_{j})+1 \end{array}$$
(19)$$\begin{array}{c} \chi_{Eve,k}^{4}=\varepsilon_{1}\sum_{j=1}^{K}\mathrm{Tr}(F_{Eve}W_{j})+\gamma_{1}\sum_{j=1}^{K}\mathrm{Tr}(F_{Eve}Z_{j})+1 \end{array}$$

However, P1 is still a difficult non-convex optimization problem to solve. For P1, the traditional method is to adopt the SCA [42]. Specifically, the wide upper bounds of $\log_{2}\left(\chi_{k}^{3}\right)$ and $\log_{2}\left(\chi_{Eve,k}^{4}\right)$ are constructed by Taylor expansion, then the partial derivatives of the upper bounds are solved, and finally the approximate value of $R_{k}^{d}-R_{Eve}^{d}$ is obtained by inequality scaling [42]. However, as the number of APs and users increases, and the dimension of the channel matrix increases, SCA needs to be used to continuously approach the optimal solution, and the computational complexity increases sharply. Therefore, the paper adopts an ingenious transformation method, using Lemma 1 [43,44] to convert the problem of logarithmic subtraction into the problem of solving the extremum of a single function, which can effectively reduce the computational complexity.

**Lemma 1.** 
*The function $\vartheta \left(t\right)=-tx+\ln(t)+1$, for any x>0, we have*

(20)
$$\begin{array}{c} -\ln(x)=\max_{t>0}\;\vartheta \left(t\right) \end{array}$$


*And the optimal solution of the above formula is $t=\frac{1}{x}$.*


From Lemma 1, we can find the upper bound of function $\vartheta \left(t\right)$, and when $t=\frac{1}{x}$, the upper bound is tight. With Lemma 1, maximizing the user’s minimum achievable secrecy rate can be transformed into solving the maximization problem of $-\log_{2}\left(\chi_{k}^{3}\right)-\log_{2}\left(\chi_{Eve,k}^{4}\right)$. We set $x=\varepsilon_{0}\sum_{j=1,j\neq k}^{K}\mathrm{Tr}(F_{k}W_{j})+\gamma_{0}\sum_{j=1}^{K}\mathrm{Tr}(F_{k}Z_{j})+1$ and $t=t_{k}$, restate $R_{k}^{d}$ and $R_{Eve}^{d}$.
(21)$$\begin{array}{c} \begin{array}{c} \begin{array}{cc} R_{k}^{d}\ln2 & =\ln\left(\chi_{k}^{1}\right)-\ln\left(\chi_{k}^{3}\right)\\ \; & =\max_{t_{k>0}}\vartheta_{k}\left(W,Z,t_{k}\right) \end{array} \end{array} \end{array}$$
where
(22)$$\begin{array}{c} \begin{array}{c} \begin{array}{cc} R_{Eve}^{k}\ln2 & =\ln\left(\chi_{Eve,k}^{3}\right)-\ln\left(\chi_{Eve,k}^{4}\right)\\ \; & =\min_{t_{Eve}>0}\vartheta_{Eve,k}\left(W,Z,t_{Eve,k}\right) \end{array} \end{array} \end{array}$$

Treating $R_{Eve}^{d}$ in the same way, let $x=\varepsilon_{0}\sum_{j=1}^{K}\mathrm{Tr}(F_{Eve}W_{j})+\gamma_{0}\sum_{j=1}^{K}\mathrm{Tr}(F_{Eve}Z_{j})+1$ and $t=t_{Eve}$. Then, $R_{Eve}^{d}$ can be written as
(23)$$\begin{array}{c} \begin{array}{c} \begin{array}{cc} R_{Eve}^{k}\ln2 & =\ln\left(\chi_{Eve,k}^{3}\right)-\ln\left(\chi_{Eve,k}^{4}\right)\\ \; & =\min_{t_{Eve}>0}\vartheta_{Eve,k}\left(W,Z,t_{Eve,k}\right) \end{array} \end{array} \end{array}$$
where
(24)$$\begin{array}{c} \begin{array}{c} \begin{array}{cc} \vartheta_{Eve,k}\left(W,Z,t_{Eve,k}\right) & =t_{Eve,k}\chi_{Eve,k}^{3}-\ln\left(\chi_{Eve,k}^{4}\right)\\ \; & -\ln\left(t_{Eve,k}\right)-1 \end{array} \end{array} \end{array}$$

According to Sion’s min–max theory in [44], P1 can be reformulated as P1a
(25)$$\begin{array}{cc} P1a: & \max_{W,Z,t_{k},t_{Eve,k}}(\vartheta_{k}\left(W,Z,t_{k}\right)\\ & -\max_{k}\;\vartheta_{Eve,k}\left(W,Z,t_{Eve,k}\right))\\ s.t. & C1:Tr\left(W_{k}\right)+Tr\left(Z_{k})\right)⩽P_{\max},\\ & W_{k}⪰0,\;Z_{k}⪰0,\;k\in \left\{1,2,\ldots ,K\right\}.\\ & C1^{\prime }:\;t_{k}>0,\;t_{Eve,k}>0. \end{array}$$


So far, P1a has been transformed into a convex function optimization problem for W, Z, $t_{k}$ and $t_{Eve,k}$. We use alternating optimization to find the respective optimal values.

#### 4.1.1. Fix W and Z, Solve for $t_{k}$ and $t_{Eve,k}$


From Lemma 1, given W and Z, the optimal closed-form expressions for $t_{k}$ and $t_{Eve,k}$ can be written as
(26)$$\begin{array}{c} \begin{array}{c} t_{k}^{*}=\frac{1}{\gamma_{0}\sum_{j=1,j\neq k}^{K}\mathrm{Tr}(F_{k}W_{j})+\gamma_{0}\sum_{j=1}^{K}\mathrm{Tr}(F_{k}Z_{j})+1} \end{array} \end{array}$$
(27)$$\begin{array}{c} \begin{array}{c} t_{Eve,k}^{*}=\frac{1}{\gamma_{0}\sum_{j=1}^{K}\mathrm{Tr}(F_{Eve}W_{j})+\gamma_{0}\sum_{j=1}^{K}\mathrm{Tr}(F_{Eve}Z_{j})+1} \end{array} \end{array}$$

#### 4.1.2. Fix $t_{k}^{*}$, $t_{Eve,k}^{*}$, Solve W and Z

Given $t_{k}^{*}$, $t_{Eve,k}^{*}$, the optimal W and Z are solved by optimizing P1b
(28)$$\begin{array}{cc} P1b: & \max_{W,Z}\;\left\{\vartheta_{k}\left(W,Z,t_{k}^{*}\right)-\underset{k}{\max\;}\vartheta_{Eve,k}\left(W,Z,t_{Eve,k}^{*}\right)\right\}\\ s.t.\; & C1:Tr\left(W_{k}\right)+Tr\left(Z_{k}\right)⩽P_{\max}\\ & W_{k}⪰0,Z_{k}⪰0,k\in \left\{1,2,\ldots ,K\right\} \end{array}$$

Introducing auxiliary variable τ, P1b can continue to be written as P1c
(29)$$\begin{array}{cc} P1c: & \max_{W,Z,\tau }\vartheta_{k}\left(W,Z,t_{k}^{*}\right)-\tau \\ s.t. & C1:Tr\left(W_{k}\right)+Tr\left(Z_{k}\right)⩽P_{\max}\\ & W_{k}0,Z_{k}⪰0,k\in \left\{1,2,\ldots ,K\right\}\\ & C1^{\prime \prime }:\vartheta_{Eve,k}\left(W,Z,t_{Eve,k}^{*}\right)<\tau \end{array}$$

Obviously, P1c is a solvable convex form that can be solved by convex optimization toolboxes such as CVX. In C1, the rank-1 constraints are removed by semi-definite relaxation (SDR), so the solved ones are not guaranteed to be rank-1 in P1c. If the rank of W and Z in the result is 1, ${\tilde{w}}_{k}$ and ${\tilde{z}}_{k}$ can be recovered by eigenvalue decomposition; if the rank is not 1, it needs to be recovered by Gaussian randomization [41].

So far, given the IRS phase shift, the process of solving the optimal precoding and AN matrix is over, and its specific steps are given in Algorithm 1.

**Algorithm 1:** Alternating Optimization for Solving W and Z  **Input:  **$P_{\max}$, $\varepsilon_{0}$, $\varepsilon_{1}$, $F_{k}$ and $F_{Eve}$.
  **Output:** Beamforming W, AN mattrix Z.
  1: Initialize $W^{\left(0\right)}$, and $Z^{\left(0\right)}$, satisfy constraint C1, i=1.    2: **repeat**    3:   Given $W^{\left(i-1\right)}$, $Z^{\left(i-1\right)}$ according to formulation (26) and (27) get the optimal $t_{k}^{(i)*}$, $t_{Eve}^{(i)*}$.    4:   With given $t_{k}^{(i)*}$, $t_{Eve}^{(i)*}$, solving P3 get the optimal $W^{\left(i\right)}$ and $Z^{\left(i\right)}$.  5:   **Update**
i=i+1.    6: **until** the value of P3 reaches convergence.

### 4.2. Sub-Problem for $\tilde{\Theta }$ When Given W and Z


This subsection presents the solution for optimizing the IRS reflection phase with fixed W and Z. First, we rewrite the corresponding optimization objective
(30)$$\begin{array}{c} \begin{array}{c} \begin{array}{cc} P2: & \max_{\tilde{\Theta }}\min_{k}R_{\sec,k}^{d}\\ s.t. & C2:0⩽\theta_{n}⩽2\pi ,\left|e^{j\theta_{n}}\right|\;=\;1,\;n\in \left\{1,\ldots ,N\right\} \end{array} \end{array} \end{array}$$

Similar to the solution of (IV-A), we first make some deformations of the objective function and then apply Lemma 1 to solve it. For convenience, we define $\lambda_{i}={\left[e^{j\theta_{i,1}},e^{j\theta_{i,2}},\ldots e^{j\theta_{i,N}}\right]}^{H}$, $\zeta =\left[1,\lambda_{1}^{H},\lambda_{2}^{H},\ldots ,\lambda_{I}^{H}\right]$, $\zeta \in C^{1\times (IN+1)}$.
(31)$$\begin{array}{c} \begin{array}{c} \begin{array}{c} G_{m,j,l}=\left[\begin{array}{c} h_{m,l}^{H}w_{j,l}\\ diag\left(g_{m,1}^{H}\right)G_{1,l}w_{j,l}\\ diag\left(g_{m,2}^{H}\right)G_{2,l}w_{j,l}\\ \ldots \\ diag\left(g_{m,I}^{H}\right)G_{I,l}w_{j,l} \end{array}\right] \end{array} \end{array} \end{array}$$
(32)$$\begin{array}{c} \begin{array}{c} \begin{array}{c} T_{m,j,l}=\left[\begin{array}{c} h_{m,l}^{H}z_{j,l}\\ diag\left(g_{m,1}^{H}\right)G_{1,l}z_{j,l}\\ diag\left(g_{m,2}^{H}\right)G_{2,l}z_{j,l}\\ \ldots \\ diag\left(g_{m,I}^{H}\right)G_{I,l}z_{j,l} \end{array}\right] \end{array} \end{array} \end{array}$$
(33)$$\begin{array}{c} \begin{array}{c} \begin{array}{c} V_{k,j,Ja}=\left[\begin{array}{c} h_{k,Ja}^{H}w_{j,Ja}\\ diag\left(g_{k,1}^{H}\right)G_{1,Ja}w_{j,Ja}\\ diag\left(g_{k,2}^{H}\right)G_{2,Ja}w_{j,Ja}\\ \ldots \\ diag\left(g_{k,I}^{H}\right)G_{I,Ja}w_{j,Ja} \end{array}\right] \end{array} \end{array} \end{array}$$
where $m\in \left\{1,2,\ldots K,Eve\right\}$. Substituting ζ, G, T and V into Formulas (2) and (3), the information received by the user and Eve is rewritten as
(34)$$\begin{array}{c} \begin{array}{c} \begin{array}{c} y_{k}=\sum_{j=1}^{K}\sum_{l\in M_{j}}\zeta G_{k,j,l}+\sum_{j=1}^{K}\sum_{l\in M_{j}}\zeta T_{k,j,l}+\sum_{j=1}^{K}\zeta V_{k,j,Ja}+n_{k} \end{array} \end{array} \end{array}$$
(35)$$\begin{array}{c} \begin{array}{c} \begin{array}{c} y_{Eve}^{k}=\sum_{j=1}^{K}\sum_{l\in M_{j}}\zeta G_{Eve,j,l}+\sum_{j=1}^{K}\sum_{l\in M_{j}}\zeta T_{Eve,j,l}+n_{Eve} \end{array} \end{array} \end{array}$$

Define $T=\zeta^{H}\zeta$, $R_{m,j,l}=G_{m,j,l}G_{m,j,l}^{H}$, $P_{m,j,l}=T_{m,j,l}T_{m,j,l}^{H}$, $B_{k,j,Ja}=V_{k,j,Ja}V_{k,j,Ja}^{H}$. Then, the achievable secrecy rate of user *k* can be written as formulation (36).
(36)$$\begin{array}{c} \begin{array}{c} \begin{array}{cc} R_{k}^{d}-R_{Eve,k}^{d} & =\log_{2}\left(\gamma_{k}^{1}\right)+\log_{2}\left(\gamma_{Eve,k}^{2}\right)\\ \; & -\log_{2}\left(\gamma_{k}^{3}\right)-\log_{2}\left(\gamma_{Eve,k}^{4}\right) \end{array} \end{array} \end{array}$$
where
(37)$$\begin{array}{c} \begin{array}{c} \begin{array}{cc} \gamma_{k}^{1} & =\varepsilon_{0}\sum_{j=1}^{K}\sum_{l\in M_{j}}\mathrm{Tr}(R_{k,j,l}T)+\varepsilon_{0}\sum_{j=1}^{K}\sum_{l\in M_{j}}\mathrm{Tr}(P_{k,j,l}T)\\ & +\sum_{j=1}^{K}\mathrm{Tr}(B_{k,j,Ja}T)+1 \end{array} \end{array} \end{array}$$
(38)$$\begin{array}{c} \begin{array}{c} \begin{array}{cc} \gamma_{Eve,k}^{2} & =\varepsilon_{0}\sum_{j=1,j\neq k}^{K}\sum_{l\in M_{j}}\mathrm{Tr}(R_{Eve,j,l}T)\\ \; & +\varepsilon_{0}\sum_{j=1}^{K}\sum_{l\in M_{j}}\mathrm{Tr}(P_{Eve,j,l}T)+1 \end{array} \end{array} \end{array}$$
(39)$$\begin{array}{c} \begin{array}{c} \begin{array}{cc} \gamma_{k}^{3} & =\varepsilon_{0}\sum_{j=1,j\neq k}^{K}\sum_{l\in M_{j}}\mathrm{Tr}(R_{k,j,l}T)+\varepsilon_{0}\sum_{j=1}^{K}\sum_{l\in M_{j}}\mathrm{Tr}(P_{k,j,l}T)\\ \; & +\sum_{j=1}^{K}\mathrm{Tr}(B_{k,j,Ja}T)+1 \end{array} \end{array} \end{array}$$
(40)$$\begin{array}{c} \begin{array}{c} \begin{array}{cc} \gamma_{Eve,k}^{4} & =\varepsilon_{0}\sum_{j=1}^{K}\sum_{l\in M_{j}}\mathrm{Tr}(R_{Eve,j,l}T)\\ \; & +\varepsilon_{0}\sum_{j=1}^{K}\sum_{l\in M_{j}}\mathrm{Tr}(P_{Eve,j,l}T)+1 \end{array} \end{array} \end{array}$$

Similar to beamforming and the solution of AN, with W and Z fixed, we use Lemma 1 and SDR to solve the optimal ζ, and the optimization problem can be formulated as
(41)$$\begin{array}{cc} P2a: & \max_{T,\partial_{k},\partial_{Eve,k}}\left\{{℘}_{k}\left(T,\partial_{k}\right)-\max_{k}\;{℘}_{Eve,k}\left(T,\partial_{Eve,k}\right)\right\}\\ s.t. & C3:T⪰0,T_{n,n}=1,\;n=1,\ldots ,IN+1.\\ & C4:\partial_{k}>0,\partial_{Eve,k}>0 \end{array}$$

P2a is a coupled optimization about T and $\left(\partial_{k},\partial_{Eve,k}\right)$. When one of the two is fixed, the other variable is convex, which can be solved directly using CVX. Fix T, and the optimal $\left(\partial_{k},\partial_{Eve,k}\right)$ can be solved by the following formula $\partial_{k}^{*}=\frac{1}{\gamma_{k}^{3}}$ and $\partial_{Eve,k}^{*}=\frac{1}{\gamma_{Eve,k}^{4}}$.

Given $\left(\partial_{k}^{*},\partial_{Eve,k}^{*}\right)$, we obtain the optimal T by solving P2b.
(42)$$\begin{array}{c} \begin{array}{c} \begin{array}{cc} P2b: & \max_{T_{n,n}=1}\left\{{℘}_{k}\left(T,\partial_{k}^{*}\right)-\max_{k}\;{℘}_{Eve,k}\left(T,\partial_{Eve,k}^{*}\right)\right\}\\ s.t.\;\; & C3. \end{array} \end{array} \end{array}$$

Finally, λ is recovered by eigenvalue decomposition and Gaussian randomization. So far, the beamforming W, the AN Z matrix and the IRS reflection phase $\tilde{\Theta }$ are solved, and the overall solution is given in Algorithm 2.

**Algorithm 2:** BCD-based algorithm for solving P1**Input:** Number of AP *L*, number of IRS *I*, number of *N*, number of antennas *Q*, $P_{\max}$, $\varepsilon_{0}$, $\varepsilon_{1}$$F_{k}$ and $F_{Eve}$, $H_{k,Ja}$, ϖ, aa.
**Output:** Beamforming W, AN mattrix Z, phase-shift matrix $\tilde{\Theta }$.
1:  **Initialize** phase-shift matrix ${\tilde{\Theta }}^{\left(0\right)}$, m=1. 2:  **repeat**  3:  Solve each sub-problem: given ${\tilde{\Theta }}^{\left(m-1\right)}$, solve (P1a) by applying Algorithm 1, get the solution $W^{\left(m\right)}$, $Z^{\left(m\right)}$; solve (p2a) for given $W^{\left(m\right)}$, $Z^{\left(m\right)}$ and obtain the solution ${\tilde{\Theta }}^{\left(m\right)}$.4:  **Update**
m=m+1. 5:  **until** the value of P1 is below ϖ or m=aa.


## 5. Numerical Results and Discussion

As shown in Figure 3, in the simulation of IRS-assisted cell-free massive MIMO secure transmission, we build the simulation scenario with reference to the deployment scheme of [13,14,41]. We set up five APs in the x-z plane with each AP height set to 8 m and the APs spaced 30 m apart. Multiple IRSs are randomly located within a circle of 5 m radius at a height of 4 m in the y-axis direction 15 m from the base station. To investigate the worst-case potential of the system to resist transient jamming and eavesdropping, we set up SJD to lurk within a cluster of users, who are randomly distributed within a square area with a 10 m side at the central coordinates (55, 15, 0). Considering that IRS location distribution has a significant impact on transmission performance, higher gain can be obtained by deploying close to the base station [22,28]. Therefore, to investigate the upper bound of secure transmission performance for multi-IRS-assisted cell-free massive MIMO systems, we set the IRS close to the AP rather than the user and consider that the CSI of each channel is known at the CPU [32]. As in Figure 3, it is assumed that there are obstacles between the AP and user areas that cause large AP and user channel fading. Specifically, it is assumed that AP-Users/Eve, AP/Jammer-IRS, and Jammer-Users all follow a distance-dependent path loss model. Take the direct channel between AP *l* and user *k* as an example.
(43)$$\begin{array}{c} \begin{array}{c} \begin{array}{c} h=\sqrt{L_{0}d_{AU}^{-\delta_{AU}}}h^{*} \end{array} \end{array} \end{array}$$
(44)$$\begin{array}{c} \begin{array}{c} \begin{array}{c} h^{*}=\sqrt{\frac{\kappa_{AU}}{\kappa_{AU}+1}}h^{LoS}+\sqrt{\frac{1}{1+\kappa_{AU}}}h^{NLoS} \end{array} \end{array} \end{array}$$
where $L_{0}=-40\;\mathrm{dB}$ represents the path loss when the reference distance is 1m, $d_{AU}$ represents the distance between the AP and the user, and $\delta_{AU}$ represents the path loss exponent between the AP and the user. $h^{*}$ represents the small-scale fading component, which is modeled as a Rician fading model [22,28,43]. $\kappa_{AU}$ represents the Rician factor of the AP and the user, and $h^{LoS}$ and $h^{NLoS}$ denote the line-of-sight (LoS) component and the non-line-of-sight (NLoS) component, respectively. Other parameters are shown in Table 1.

In Figure 4, the achievable secrecy rates of the various schemes are compared versus IRS reflection amplitude. As illustrated in Figure 4, an increase in the reflection amplitude is associated with a corresponding rise in the minimum achievable secrecy rate under each scheme. For a given reflection amplitude, the proposed scheme of multiple IRSs combined with AN is effective in increasing the minimum achievable secrecy rate by approximately 3% compared to a single IRS and AN-assisted scheme. The proposed multiple IRS and AN scheme improves the achievable secrecy rate by 10% compared to a single IRS-assisted scheme without AN. In addition, the number of IRSs has a greater impact on the confidentiality performance of the system than AN. As the number of IRSs increases, the number of cascaded channels between the AP and users increases, resulting in higher channel gains. Although the number of cascaded channels between interferers and users has also increased, it should be noted that the reflected phase and transmitted precoding of multiple IRSs are jointly optimized based on user information (such as location) and demand, and they are reliable for users. Interferers cannot use cascaded channels to obtain sufficient signal interference, and this phenomenon is more pronounced when the number of IRSs increases.

Figure 5 compares the curves of achievable secrecy rate vs. transmission power obtained under various schemes. As can be seen from the figure, as the transmission power increases, the system’s achievable secure rate also increases. This is in line with the fact that as the transmission power increases, the user end receives a stronger signal, the ability to resist instantaneous interference and eavesdropping is enhanced, and the system’s secure rate increases. The fixed transmission power is 30 dBm, the transmission scheme jointly designed by multiple IRSs and AN can improve the user secrecy rate by about 18% compared with a single IRS. Continuing to increase the transmission power, the performance of the multi-IRS scheme without AN is close to that of the proposed scheme, because at this time, the transmission power is large enough, and the channel capacity of the legitimate user is completely greater than that of the Eve. It is not a wise choice to use the base station to redistribute power for designing AN.

Figure 6 illustrates the minimum secure rate that can be achieved for each scheme at varying transmission powers of the jammer. From the graph, the various schemes using IRS against transient interference achieve better performance when the jammer’s power is small. As the interference power increases, the achievable confidential rate obtained by the scheme of a single IRS is found to be significantly lower than that of the scheme proposed in this paper, and it is also lower than that of the multi-IRS scheme without AN transmission. This is due to the fact that the multi-IRS provides users with reliable channel gain through cascaded channels, thus ensuring the normal downlink communication quality under high-power interference. In general, the auxiliary scheme with multiple IRSs is better than the single IRS, and the auxiliary scheme with AN is better than the scheme without AN when the jammer’s power is low.

Figure 7 shows the achievable secrecy rate versus the number of users *K* for the different schemes. It can be seen from Figure 7 that the secrecy rate achieved by each scheme decreases as the number of users increases, which is limited by the total transmission power. The proposed scheme with multiple IRSs and AN performs better than the scheme with a single IRS and AN for different numbers of users, and it performs slightly better than the scheme with multiple IRSs but without AN. However, this superiority decreases as the number of users increases, because the increase in the number of users reduces the power allocated to each user, but the Eve and jammer power remain unchanged, and the gain of AN with limited power is difficult to achieve to effectively reduce the channel capacity of Eve. Therefore, the proposed scheme is more favorable for a smaller number of users.

Figure 8 illustrates the relationship between the system’s achievable secrecy rate and the location of the IRS cluster. Here, Setup(a), Setup(b), and Setup(c) represent the three scenarios where the IRS cluster is located close to the AP cluster, in the middle of the AP and the user, and close to the user cluster, respectively. From the diagram, it can be seen that higher secrecy capacity can be achieved when the IRS cluster is close to the AP, followed by deploying the IRS cluster in the middle of the AP and the user, and worst of all, when the IRS is deployed close to the user. This is because when the user and AP are far away, it is the IRS cascaded link that plays the main role, and when the IRS cluster is deployed near the AP, higher transmit power can be allocated, and the high channel gain of the cascaded link improves the channel capacity for the user. Because the SJD is deployed in the middle of the user cluster, when the IRS cluster is close to the user, the SJD can also take advantage of the IRS reflection gain, increasing the interference to legitimate users and improving the Eve’s channel capacity. The Setup(c) scenario therefore has the lowest system secrecy rate.

## 6. Conclusions

This paper studies the use of multiple IRSs and AN to enhance the anti-interference and anti-eavesdropping capabilities of cell-free massive MIMO. In order to ensure secure downlink transmission, a joint optimization algorithm for AP beamforming, AN matrix and IRS phase shift matrix is proposed based on the BCD algorithm, which maximizes the minimum secrecy rate of all users. Avoiding heavy signal processing by the CPU and huge backhaul link load, an AP selection strategy based on user location information is designed to reasonably determine the set of APs for each user so as to achieve a trade-off between signal processing complexity and security performance with the optimal AP set selection. Simulation experiments show that the proposed multi-IRS joint AN scheme is superior to the traditional single IRS scheme and slightly better than the transmission scheme without AN assistance. The scheme proposed provides ideas for secure transmission from single IRS to multi-IRS assistance in wireless networks and also broadens the application of multi-IRS in 6G networks. In the future, further consideration can be given to the extreme communication environment of multi-SJD joint interference, and the security of multi-IRS enhancement systems can be studied from the perspective of game equilibrium between SJD and users.

## Figures and Tables

**Figure 1 sensors-24-07326-f001:**
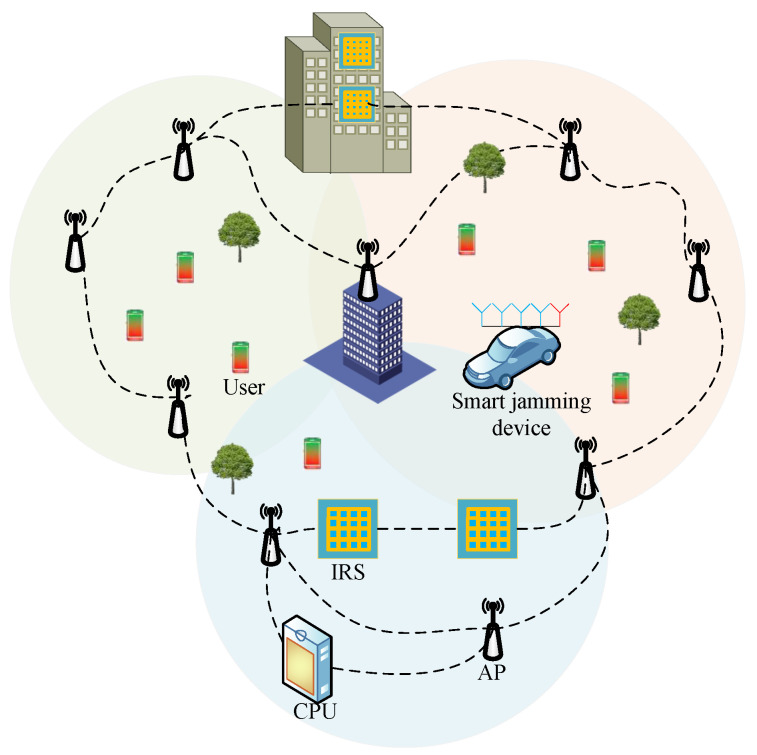
Schematic diagram of multi-IRS assisted communication.

**Figure 2 sensors-24-07326-f002:**
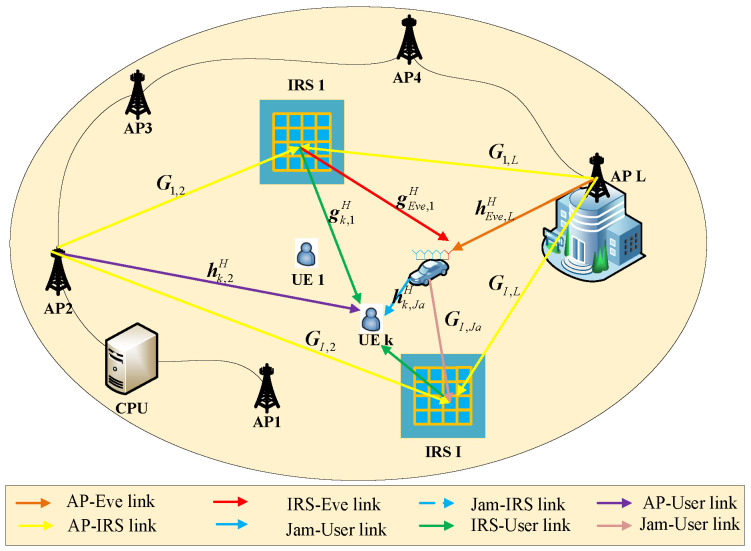
System model of multi-IRS against simultaneous jamming and eavesdropping communication.

**Figure 3 sensors-24-07326-f003:**
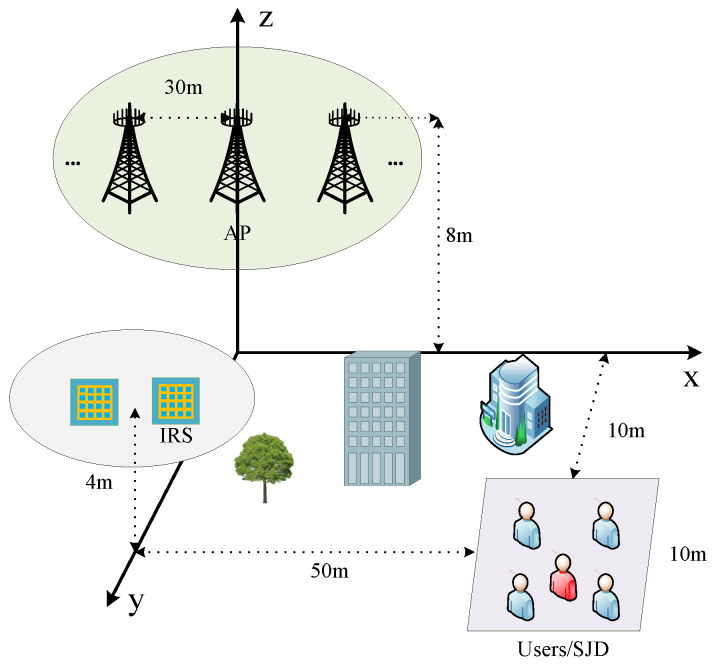
Simulation deployment.

**Figure 4 sensors-24-07326-f004:**
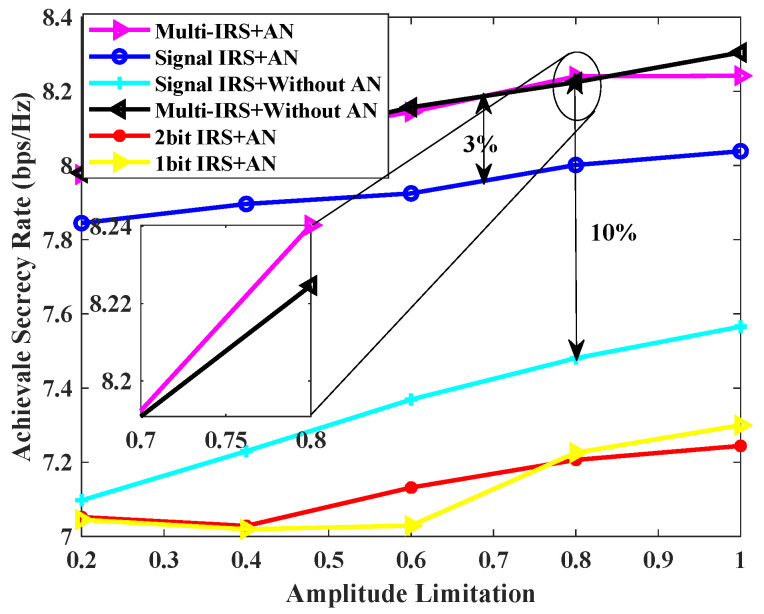
Achievable secrecy rate [bps/Hz] v.s. IRS amplitude limitation for L=5, K=2, N=20, $P_{\max}$ = 0 dBm, $P_{Ja,\max}$ = 30 dBm.

**Figure 5 sensors-24-07326-f005:**
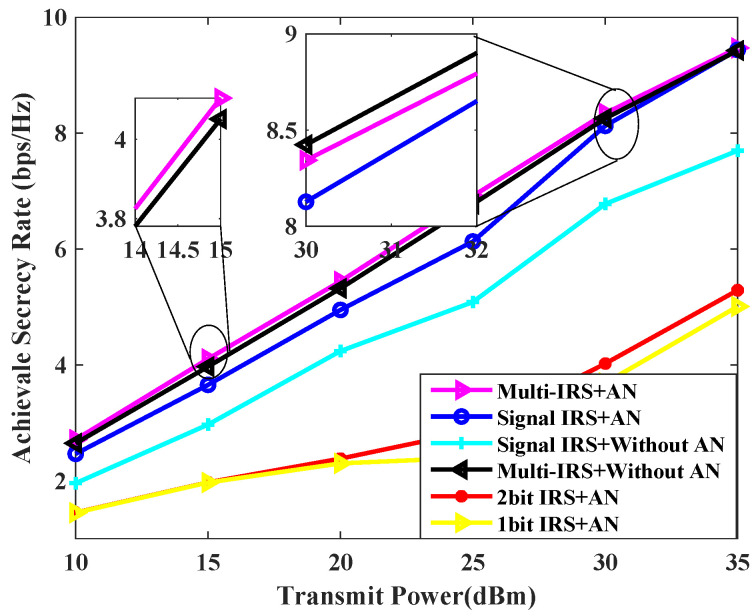
Achievable secrecy rate [bps/Hz] v.s. transmit power for L=5, K=2, N=20, $P_{Ja,\max}$ = 30 dBm.

**Figure 6 sensors-24-07326-f006:**
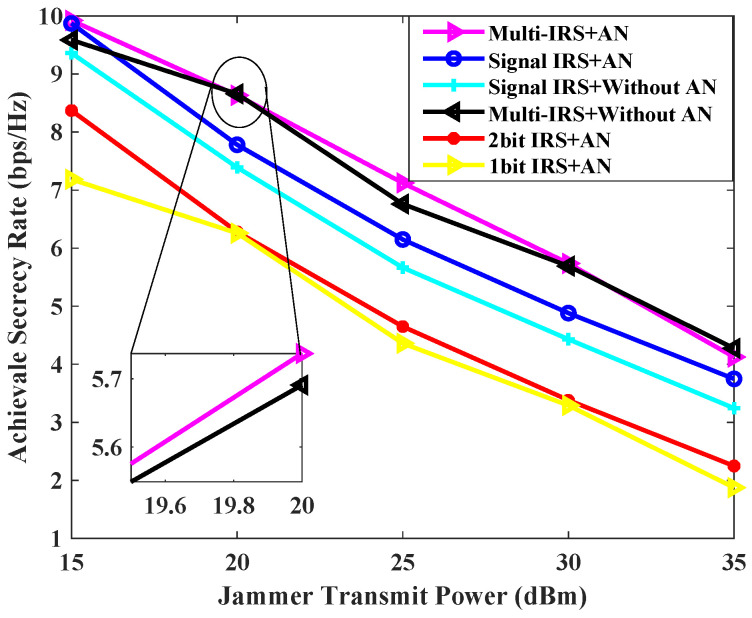
Achievable secrecy rate [bps/Hz] v.s. jammer transmit power [dBm] for L=5, K=2, N=20, $P_{\max}$ = 0 dBm.

**Figure 7 sensors-24-07326-f007:**
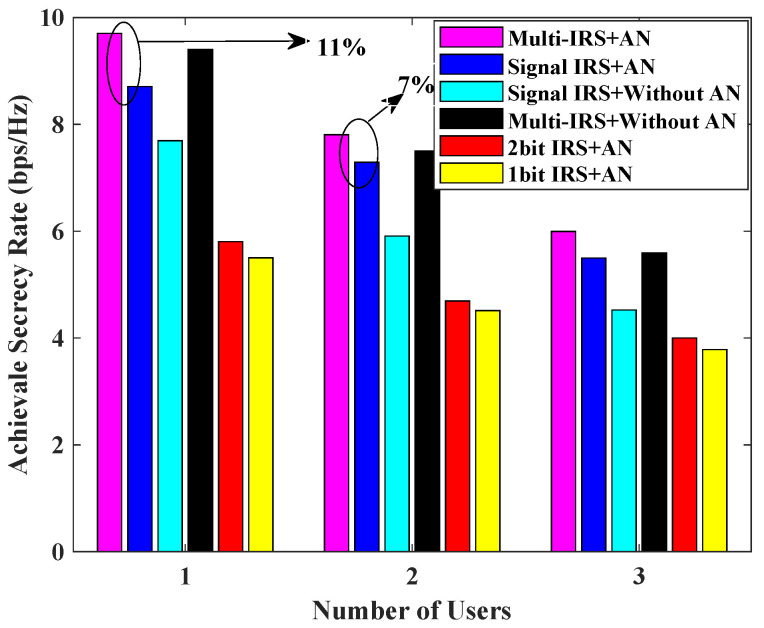
Achievable secrecy rate [bps/Hz] v.s. number of users for L=5, N=20, $P_{\max}$ = 0 dBm, $P_{Ja,\max}$ = 30 dBm.

**Figure 8 sensors-24-07326-f008:**
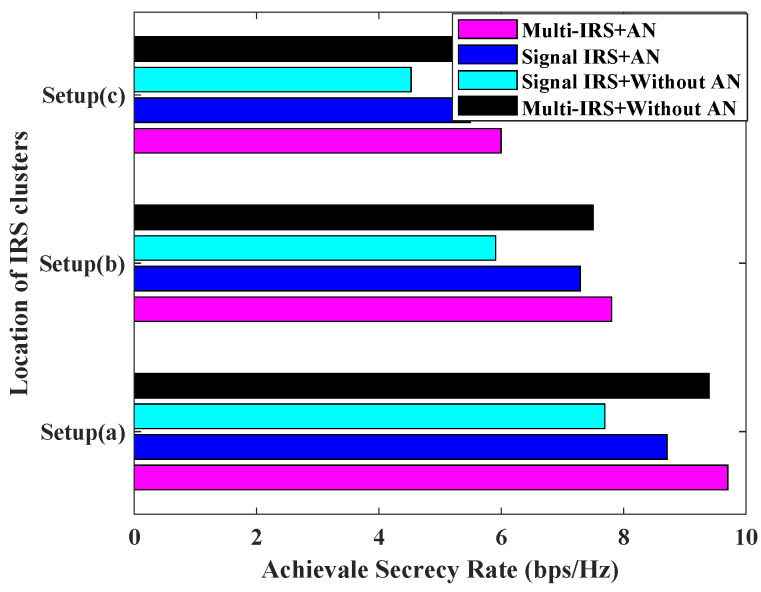
Achievable secrecy rate [bps/Hz] v.s. location of IRS clusters for L=5, K=2, N=20, $P_{\max}$ = 0 dBm, $P_{Ja,\max}$ = 30 dBm.

**Table 1 sensors-24-07326-t001:** List of key notations.

Parameter	Value
Bandwidth	180 MHz
$P_{\max}$	30 dBm
$P_{Ja,\max}$	30 dBm
Noise power of user/Eve	−80 dBm
Antennas of each AP and SJD	Q = $Q_{e}$ = 2
Path loss (AP/jammer–IRS)	$\delta_{AR}=\delta_{JR}=2.2$
Path loss (AP–user/Eve)	$\delta_{AU}=\delta_{AE}=3.5$
Path loss (IRS–user/Eve)	$\delta_{RU}=\delta_{RE}=3$
Rician factor (AP/jammer–IRS)	$\kappa_{AR}=\kappa_{JR}=\infty$
Rician factor (AP/IRS–user/Eve)	$\kappa_{AU}=\kappa_{AE}=\kappa_{RU}=\kappa_{RE}=0$

## Data Availability

The original contributions presented in the study are included in the article, further inquiries can be directed to the corresponding author.

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
