# Peer review of "Multi-Intelligent Reflecting Surfaces and Artificial Noise-Assisted Cell-Free Massive MIMO Against Simultaneous Jamming and Eavesdropping"

_sensors, 2024, doi:10.3390/s24227326_

Round 1

Reviewer 1 Report

Comments and Suggestions for Authors

1. Clarity: The abstract briefly introduces the research content and methods, but it is recommended to highlight the innovation of the research. For example, why is this research superior to existing methods in combining intelligent reflecting surfaces (IRS) and artificial noise (AN)? Its superiority can be emphasized through quantitative results (for example, the specific percentage of security rate improvement).

Background explanation: Although the research background is briefly described, the importance of this problem in practical applications can be more clearly explained, especially for potential application scenarios in 6G networks.

2. Background and motivation: The introduction has already outlined the challenges of physical layer security and massive MIMO, but the further description of "smart jamming devices" (SJDs) should be strengthened. Its potential threat to system security and why existing technologies are not sufficient to meet this challenge can be explained in more detail.

Related literature: In terms of the latest progress of IRS technology, although some literature is cited, more practical application cases and challenges in the multi-intelligent reflecting surface (multi-IRS) environment can be added. In addition, consider adding a discussion of other anti-interference and eavesdropping schemes to highlight the uniqueness of this research.

3. Symbols and definitions: In complex mathematical derivations, the definitions of symbols need to be clearer. Although a large number of symbols are used, the physical meaning of some symbols is not fully explained in the context, such as some variables in Section 2. It is recommended to add an explanation of their physical meaning when these symbols are first introduced.

Optimization framework: For the block coordinate descent (BCD) algorithm and semidefinite relaxation (SDR) method used, although the methods are appropriate, the complexity of the optimization and the specific computational cost are not fully discussed for readers. It is recommended to add a quantitative description of the computational time and complexity analysis, especially the computational efficiency of the algorithm in large-scale systems, in the algorithm section.

4. Realism of the system model: In the hypothesis section, the possibility of IRS in actual deployment can be further explored, including the complexity of its hardware implementation, energy consumption, etc. Although the system model is based on ideal channel state information (CSI), CSI may not be completely accurate in actual applications. It is recommended to consider the situation of imperfect CSI in the system model and provide countermeasures.

Reasonableness of assumptions: The assumption that the smart jammer (SJD) will not be interfered with (assuming that the self-interference cancellation technology can eliminate the interference) may be too idealistic. The degradation of system performance when self-interference cannot be completely eliminated can be explored, and countermeasures can be proposed.

5. Explanation of quantitative results: When discussing the simulation results, it is recommended to provide a more detailed physical explanation. Why is the combination of multiple IRS and AN more effective in improving the minimum achievable safety rate? In particular, the impact of the transmission power change on the system safety rate in Figure 5 needs a deeper explanation.

Comparative analysis: Although the simulation results show that the multiple IRS is superior to the single IRS solution, there is no detailed discussion of why the number of IRSs has such a large impact on performance. The role of the number of IRSs can be better explained by analyzing the relationship between the signal path and the interference path.

Chart improvements: Most charts show the performance trend of the system, but there is no error analysis or confidence interval annotation of the data. It is recommended to add statistical information of the experimental results, such as confidence intervals or error bars, to more clearly show the robustness of the results.

6. In-depth discussion: The conclusion section is relatively brief, but it can add a prospect for the future application of this research. For example, how to apply multiple IRS and AN solutions in future 6G networks? How is the scalability of this solution? In addition, how to further improve the security of the system (for example, considering the scenario of multiple intelligent jammers)?

Author Response

On behalf of my co-authors, we thank you for giving us a chance to revise and improve the quality of our article.

 We have read the reviewers’ comments carefully and have made revision which marked in red in the manuscript. We have tried our best to revise our manuscript according to the comments: “Multi-Intelligent Reflecting Surfaces and Artificial Noise-Assisted Cell-Free Massive MIMO against Simultaneous Jamming and Eavesdropping (3270186)”.

Attached please find the revision, which we hope that you will find this updated manuscript to your satisfaction and consider it for plication as an article in Journal sensors. Here is a point -by-point response to the reviewers’ comments and concerns. The reviewer's comments are listed below in black text, with the specific issues numbered. We have provided a clear and detailed response in blue text, and have made the necessary changes/additions to the manuscript in red bold italic text.

Reviewer 1:

  1. Clarity: The abstract briefly introduces the research content and methods, but it is recommended to highlight the innovation of the research. For example, why is this research superior to existing methods in combining intelligent reflecting surfaces (IRS) and artificial noise (AN)? Its superiority can be emphasized through quantitative results (for example, the specific percentage of security rate improvement).

Response: We feel great thanks for your professional review work on our manuscript. We strongly agree with your suggestion to highlight the innovation of the research in the abstract. In the abstract, we compare the user's achievable secrecy rate under different anti-jamming schemes to quantitatively demonstrate that the proposed multiple intelligent reflecting surfaces combined with artificial noise assistance outperforms existing schemes. For example, Specifically, in a single-user scenario, the scheme of multiple intelligent reflecting surfaces combined with artificial noise can improve the user's achievable secrecy rate by about 11% compared to the existing method (single intelligent reflective surface combined with artificial noise), and about 2% compared to the scheme assisted by multiple intelligent reflecting surfaces without artificial noise assistance”. The specific changes we have made are highlighted in red in the abstract section of the manuscript.

Background explanation: Although the research background is briefly described, the importance of this problem in practical applications can be more clearly explained, especially for potential application scenarios in 6G networks

Response: We sincerely appreciate the valuable comments. We have added the impact of malicious interference and eavesdropping on wireless networks in the background, and further explained the important application scenarios of anti-jamming technology in 6G networks. For example, “In wireless communication systems, interference and eavesdropping can cause serious practical harm, such as reducing communication quality, causing communication disruptions, leaking personal privacy, stealing trade secrets, hindering industrial production, and threatening national security. In the 6G network, anti-interference has many potential application scenarios, such as being used in intelligent transportation systems, the industrial Internet of Things, safeguarding military operations, protecting the core secrets of enterprises, the financial industry, and the government, and enhancing personal privacy communications”. The specific changes we have made are highlighted in red in the Background of the manuscript.

  1. Background and motivation: The introduction has already outlined the challenges of physical layer security and massive MIMO, but the further description of "smart jamming devices" (SJDs) should be strengthened. Its potential threat to system security and why existing technologies are not sufficient to meet this challenge can be explained in more detail.

Response: We appreciate the thoughtful review and constructive feedback provided by the reviewers. In the Background and Motivation, we further describe smart jamming devices and explain in detail the serious threat it poses to wireless communications and the current state of research. By citing important literature, we show that current anti-jamming methods are difficult to cope with smart jamming and there is still less research on smart jamming devices. For example, With the rapid development of artificial intelligence, smart jamming techniques have been further studied. The existing intelligent interference technology is a multiple overlapping of cognitive interference and intelligent interference in terms of connotation and extension[12]. Most scholars study only from the theory of cognitive radio and jamming strategy, and there are fewer studies on smart jamming devices (intelligent devices integrating jamming and eavesdropping). Intelligent jamming devices can masquerade as legitimate users to steal information transmitted from the base station to legitimate users, and it can also send malicious messages to interfere with the reception of legitimate users. Current anti-jamming techniques only consider scenarios of jamming or eavesdropping, and rarely study secure transmission in harsh communication environments where both jamming and eavesdropping are present”. The specific changes we have made are highlighted in red in the Background and motivation of the manuscript.

[12] PIETRO R D, OLIGERI G. Jamming mitigation in cognitive radio networks[J]. IEEE Network, vol. 27, no. 3 pp, 10-15, 2013.

Related literature: In terms of the latest progress of IRS technology, although some literature is cited, more practical application cases and challenges in the multi-intelligent reflecting surface (multi-IRS) environment can be added. In addition, consider adding a discussion of other anti-interference and eavesdropping schemes to highlight the uniqueness of this research.

Response: Thanks for your suggestion. We have checked the literature carefully and added more references on multi-IRS and against Simultaneous

Jamming and Eavesdropping into the INTRODUCTION part in the revised manuscript. For example, “The literature [24] proposes a signal beam routing methodology utilizing multiple IRS reflections, with the objective of investigating wireless power transfer from a multiple antenna base station to multiple energy users (EUs).  The proposal set forth in literature [25] is to provide support for disparate user groups through the deployment of multiple RSs. This approach effectively mitigates interference and leverages secondary reflections between IRSs, thereby enhancing the data rate for each user”. The specific alterations that have been implemented are indicated in red on the Related literature of the manuscript.

[24] W. Mei, D. Wang, Z. Chen and R. Zhang, "Joint Beam Routing and Resource Allocation Optimization for Multi-IRS-Reflection Wireless Power Transfer," in IEEE Transactions on Wireless Communications, 2024.

[25] T. V. Nguyen, D. N. Nguyen, M. D. Renzo and R. Zhang, "Leveraging Secondary Reflections and Mitigating Interference in Multi-IRS/RIS Aided Wireless Networks," in IEEE Transactions on Wireless Communications, vol. 22, no. 1, pp. 502-517, Jan. 2023.

  1. Symbols and definitions: In complex mathematical derivations, the definitions of symbols need to be clearer. Although a large number of symbols are used, the physical meaning of some symbols is not fully explained in the context, such as some variables in Section 2. It is recommended to add an explanation of their physical meaning when these symbols are first introduced.

Response: We apologize for our oversight. Thank you for your reminder. In Notation, we have added explanations for some symbols. For example, “Lowercase bold letters represent a vector, while uppercase bold letters represent a matrix. denotes the Euclidean norm of  ” . It is worth noting that the remaining symbols are all explained in detail below the formulas in the paper.

Optimization framework: For the block coordinate descent (BCD) algorithm and semidefinite relaxation (SDR) method used, although the methods are appropriate, the complexity of the optimization and the specific computational cost are not fully discussed for readers. It is recommended to add a quantitative description of the computational time and complexity analysis, especially the computational efficiency of the algorithm in large-scale systems, in the algorithm section.

Response: We feel great thanks for your professional review work on our article. We concur with the recommendation to conduct a detailed analysis of the computational complexity and computational time of the proposed algorithm. In our simulation, which was conducted on a high-performance calculator, the actual number of iterations was less than 200, and the overall simulation speed was notably rapid. Given the length of the article, we do not intend to devote a significant portion of the article to an in-depth examination of computational time and complexity.

  1. Realism of the system model: In the hypothesis section, the possibility of IRS in actual deployment can be further explored, including the complexity of its hardware implementation, energy consumption, etc. Although the system model is based on ideal channel state information (CSI), CSI may not be completely accurate in actual applications. It is recommended to consider the situation of imperfect CSI in the system model and provide countermeasures.

Response: We feel great thanks for your professional review work on our article. We fully understand the need to strengthen research on imperfect CSI. Due to graduation requirements and time constraints, this manuscript currently only considers the achievable secrecy rate under perfect CSI as the theoretical upper limit of the secrecy rate. Fortunately, we have already begun planning new experiments to fill the gap in physical layer security under imperfect CSI. We will strive to incorporate all of your suggestions in subsequent experiments.

Reasonableness of assumptions: The assumption that the smart jammer (SJD) will not be interfered with (assuming that the self-interference cancellation technology can eliminate the interference) may be too idealistic. The degradation of system performance when self-interference cannot be completely eliminated can be explored, and countermeasures can be proposed.

Response: We appreciate the thoughtful review and constructive feedback provided by the reviewers. We deeply agree with your point of view. The claim that the smart jamming device (SJD) is immune to self-interference is idealized. In this paper, only the damage to the system caused by the SJD acting as a malicious jammer is considered, and the effect of the proposed scheme on enhancing the secrecy rate is verified. For a SJD, the self-interference it suffers has little effect on the system. If its jamming effect is weakened, it can increase its destructive power by increasing the jamming power. As you can see, we are considering studying in subsequent experiments how the SJD can enhance the jamming effect when it is subject to self-interference. For the system, we are considering using game theory ideas and methods to study how to reduce the interference to the system.

  1. Explanation of quantitative results: When discussing the simulation results, it is recommended to provide a more detailed physical explanation. Why is the combination of multiple IRS and AN more effective in improving the minimum achievable safety rate? In particular, the impact of the transmission power change on the system safety rate in Figure 5 needs a deeper explanation.

Response: We feel great thanks for your professional review work on our article. First, when there is no direct link between the user and the AP or the direct link is weak, IRS can reconstruct the cascaded channel between the AP and the user, significantly improving the user's downlink communication quality. When there is an eavesdropper (Eve), the Eve disguises itself as a legitimate user and intercepts user information through the AP-Eve direct link or the AP-IRS-Eve cascaded link. The gain of a single IRS constructed channel is insufficient to compensate for the information rate stolen by Eves. Setting up multiple IRSs, jointly optimizing IRS reflection phase and transmitter precoding, and utilizing the directional gain of multiple cascaded channel gains and beamforming to eliminate the reduced secrecy rate of Eves. Considering further limiting the information rate available to the Eve, the transmitter emits noise with a lower transmission power. The artificial noise (AN) matrix is alternately optimized with precoding and IRS phase matrices to minimize interference to legitimate users and maximize the destruction of the information received by the Eve. Therefore, the joint design of multiple IRSs and AN can effectively increase the confidentiality of downlink transmissions compared to traditional IRS assistance systems.

For Fig 5, we explain in detail the impact of changes in transmission power on system secrecy rate. The details are as follows.  “Fig.5 compares the curves of achievable secrecy rate vs. transmission power obtained under various schemes. As can be seen from the figure, as the transmission power increases, the system's achievable secure rate also increases. This is in line with the fact that as the transmission power increases, the user end receives a stronger signal, the ability to resist instantaneous interference and eavesdropping is enhanced, and the system's secure rate increases. The fixed transmission power is 30 dBm, the transmission scheme jointly designed by multiple IRSs and AN can improve the user secrecy rate by about 18% compared with a single IRS. Continuing to increase the transmission power, the performance of the multi-IRS scheme without AN is close to that of the proposed scheme, because at this time the transmission power is large enough, and the channel capacity of the legitimate user is completely greater than that of the Eve. It is not a wise choice to use the base station to redistribute power for designing AN.” Details have been highlighted in the revised manuscript.

Comparative analysis: Although the simulation results show that the multiple IRS is superior to the single IRS solution, there is no detailed discussion of why the number of IRSs has such a large impact on performance. The role of the number of IRSs can be better explained by analyzing the relationship between the signal path and the interference path.

Response: We appreciate the thoughtful review and constructive feedback provided by the reviewers. As you are concerned, we analyze the relationship between the user's signal reception path and the interference path and the eavesdropper's signal reception path when the number of IRSs increases, and explain in detail the impact of the number of IRSs on the system secrecy rate.

In terms of the user channel and the Eve channel, when there is only one IRS in the system, the channel gain provided by the cascaded channel is effective. Although the Eve can also receive signals through the AP-IRS-Eve cascaded link, the IRS reflection phase is designed based on the actual user information (such as location) and requirements. The cascaded link provides minimal channel gain for the Eve. When there are multiple IRSs in the system, this difference in channel gain is even greater. Users can enjoy the channel gain brought by multiple cascaded channels, but the received SINR of Eve remains almost unchanged. Therefore, the system security rate is significantly improved. In terms of user channels and interference channels, the joint optimization of channel gains for multiple IRSs reflection phases is significantly greater than the interference information transmitted by the interferer using cascaded channels, because the reflection phase and precoding are reliable for users.

We have added a detailed description to Figure 4, as follows In addition, the number of IRSs has a greater impact on the confidentiality performance of the system than AN. As the number of IRSs increases, the number of cascaded channels between the AP and users increases, resulting in higher channel gains. Although the number of cascaded channels between interferers and users has also increased, it should be noted that the reflected phase and transmitted precoding of multiple IRSs are jointly optimized based on user information (such as location) and demand, and are reliable for users. Interferers cannot use cascaded channels to obtain sufficient signal interference, and this phenomenon is more pronounced when the number of IRSs increases”. Details have been highlighted in the revised manuscript.

Chart improvements: Most charts show the performance trend of the system, but there is no error analysis or confidence interval annotation of the data. It is recommended to add statistical information of the experimental results, such as confidence intervals or error bars, to more clearly show the robustness of the results.

Response: We feel great thanks for your professional review work on our article. With regard to the physical layer security of the IRS auxiliary system, the primary objective is to ascertain an effective method for encoding the transmitted beam, reflection phase and other parameters, with a view to achieving an enhanced security rate. Therefore, the overall simulation of the system considers the performance trends under different parameters, but does not take into account the confidence interval of the experimental results. It is evident that the results align with the actual theoretical analysis. Current research on IRS is primarily focused on studying the upper bound of performance gain and system optimization. The confidence interval of the experimental results has not been specifically studied, and this is not a necessary area of investigation.

  1. In-depth discussion: The conclusion section is relatively brief, but it can add a prospect for the future application of this research. For example, how to apply multiple IRS and AN solutions in future 6G networks? How is the scalability of this solution? In addition, how to further improve the security of the system (for example, considering the scenario of multiple intelligent jammers)?

Response: We feel great thanks for your professional review work on our article. We have re-written the Conclusions according to the Reviewers’ suggestion, the details are list below.  This paper studies the use of multiple IRSs and AN to enhance the anti-interference and anti-eavesdropping capabilities of cell-free massive MIMO. In order to ensure secure downlink transmission, a joint optimization algorithm for AP beamforming, AN matrix and IRS phase shift matrix is proposed based on the BCD algorithm, which maximizes the minimum secrecy rate of all users. Avoiding heavy signal processing by the CPU and huge backhaul link load, an AP selection strategy based on user location information is designed to reasonably determine the set of APs for each user, so as to achieve a trade-off between signal processing complexity and security performance with the optimal AP set selection. Simulation experiments show that the proposed multi-IRS joint AN scheme is superior to the traditional single IRS scheme and slightly better than the transmission scheme without AN assistance. The scheme proposed provides ideas for secure transmission from single IRS to multi-IRS assistance in wireless networks and also broadens the application of multi-IRS in 6G networks. In the future, further consideration can be given to the extreme communication environment of multi-SJD joint interference, and the security of multi-IRS enhancement systems can be studied from the perspective of game equilibrium between SJD and users. The specific changes we have made are highlighted in red in the Conclusions of the revised manuscript.

Reviewer 2 Report

Comments and Suggestions for Authors

The secure transmission scheme based on multiple intelligent reflecting surfaces (IRS) combined with artificial noise (AN) assistance is proposed. The enhancement of secure transmission has been carried out with the numerical simulations. Everything looks fine. Just some corrections in the Figure.

Author Response

On behalf of my co-authors, we thank you for giving us a chance to revise and improve the quality of our article.

 We have read the reviewers’ comments carefully and have made revision which marked in red in the manuscript. We have tried our best to revise our manuscript according to the comments: “Multi-Intelligent Reflecting Surfaces and Artificial Noise-Assisted Cell-Free Massive MIMO against Simultaneous Jamming and Eavesdropping (3270186)”.

Attached please find the revision, which we hope that you will find this updated manuscript to your satisfaction and consider it for plication as an article in Journal sensors. Here is a point -by-point response to the reviewers’ comments and concerns. The reviewer's comments are listed below in black text, with the specific issues numbered. We have provided a clear and detailed response in blue text, and have made the necessary changes/additions to the manuscript in red bold italic text.

Reviewer 2#:

The secure transmission scheme based on multiple intelligent reflecting surfaces (IRS) combined with artificial noise (AN) assistance is proposed. The enhancement of secure transmission has been carried out with the numerical simulations. Everything looks fine. Just some corrections in the Figure.

Response: We appreciate the thoughtful review and constructive feedback provided by the reviewers. We have addressed all of your concerns and made extensive corrections to the previous manuscript. Despite the limited time available, we have improved the quality of the manuscript to the best of our ability. However, due to time constraints and other factors, there may still be some deficiencies in our work. We will continue to revise it in the follow-up to meet the requirements of the publishing house.

Reviewer 3 Report

Comments and Suggestions for Authors

1. How does the proposed large-scale fading-based AP selection strategy impact the overall system complexity and computational efficiency compared to other user grouping methods in cell-free massive MIMO systems?

2. How robust is the proposed scheme to imperfect channel state information (CSI) at the CPU, particularly in scenarios with dynamic user mobility or time-varying channel conditions?

3. How does the assumption of having "rough location information" impact the effectiveness of the AP selection strategy? What are the potential consequences if the location information is significantly inaccurate?

4. What is the expected gap between the suboptimal solutions obtained through the BCD algorithm and the true optimal solution of the original problem, particularly in terms of performance metrics like the minimum achievable secrecy rate?

5. How does the proposed optimization problem guarantee fairness among legitimate users in terms of their achievable secrecy rates? Maximizing the minimum achievable secrecy rate might not necessarily ensure fairness among all users. Some users might experience significantly lower secrecy rates compared to others, leading to potential dissatisfaction and security vulnerabilities.

6. How does the proposed BCD algorithm with function transformation compare to other existing methods for solving non-convex optimization problems in cell-free massive MIMO systems, particularly in terms of convergence speed, accuracy, and computational complexity?

7. Can you elaborate on the derivation of equation (13)? Specifically, how is the logarithm subtraction transformed into a single function maximization problem using Lemma 1?

Comments on the Quality of English Language

Author Response

On behalf of my co-authors, we thank you for giving us a chance to revise and improve the quality of our article.

 We have read the reviewers’ comments carefully and have made revision which marked in red in the manuscript. We have tried our best to revise our manuscript according to the comments: “Multi-Intelligent Reflecting Surfaces and Artificial Noise-Assisted Cell-Free Massive MIMO against Simultaneous Jamming and Eavesdropping (3270186)”.

Attached please find the revision, which we hope that you will find this updated manuscript to your satisfaction and consider it for plication as an article in Journal sensors. Here is a point -by-point response to the reviewers’ comments and concerns. The reviewer's comments are listed below in black text, with the specific issues numbered. We have provided a clear and detailed response in blue text, and have made the necessary changes/additions to the manuscript in red bold italic text.

Reviewer 3#:

  1. How does the proposed large-scale fading-based AP selection strategy impact the overall system complexity and computational efficiency compared to other user grouping methods in cell-free massive MIMO systems?

Response: We appreciate the thoughtful review and constructive feedback provided by the reviewers. In cell-free massive MIMO, a user-centric (UC) approach is employed, whereby a defined set of access points (APs) serves a specified number of users. In order to enhance system performance, it is necessary to implement an appropriate clustering scheme for the grouping of users, with the objective of increasing spectral efficiency and reducing the bit error rate. The most commonly employed methods are k-means and k-means++. In comparison to these forms of unsupervised machine learning (ML) algorithms, the proposed AP selection strategy, which is based on large-scale fading, ranks according to large-scale fading values and selects the number of APs that meet the communication rate of the user. The strategy selects an AP number that is neither redundant nor too small, effectively reducing the dimension of the transmission matrix, significantly reducing the computational complexity, ensuring the user's quality of service, and maintaining the user's information leakage rate within an acceptable threshold.

  1. How robust is the proposed scheme to imperfect channel state information (CSI) at the CPU, particularly in scenarios with dynamic user mobility or time-varying channel conditions?

Response: We feel great thanks for your professional review work on our manuscript. This paper puts forth a multi-IRS joint AN design with the objective of optimizing the minimum achievable secrecy rate for users in a malicious communication environment characterized by interference and eavesdropping. Setting the perfect CSI is to study the upper bound of the performance of the multi-IRS joint AN assisted secure transmission scheme based on the optimal channel peripheral conditions. For imperfect CSI, the secrecy of the system is guaranteed to be less than this upper bound. Due to time constraints, this paper only considers transmission schemes under perfect CSI. Fortunately, we have already carried out research on enhancing the system's resistance to instantaneous interference and eavesdropping under imperfect CSI, especially reliable transmission in highly mobile scenarios with time-varying channels and mobile IRSs.

  1. How does the assumption of having "rough location information" impact the effectiveness of the AP selection strategy? What are the potential consequences if the location information is significantly inaccurate?

Response: We feel great thanks for your professional review work on our article. The AP selection strategy proposed in this paper is based on large-scale fading, as can be seen from Equation (1).  By obtaining prior coarse location information (such as distance, direction, and angle), a cluster of location information is constructed for the user, and the optimal set of APs is determined using the selection strategy in Equation (1). In the event that the location information is of an insufficiently precise nature, characterized by considerable distance and angle errors, it may result in users making redundant or insufficient AP selections, thereby increasing the complexity of system calculations and hindering the establishment of an effective AP-IRS-User link. This ultimately impairs the prevention of interference and eavesdropping.

  1. What is the expected gap between the suboptimal solutions obtained through the BCD algorithm and the true optimal solution of the original problem, particularly in terms of performance metrics like the minimum achievable secrecy rate?

Response: We appreciate the thoughtful review and constructive feedback provided by the reviewers. In the context of convex optimization problems, such as that represented by (P1), the traditional solution method is to employ successive convex approximation. This entails the construction of a wide upper bound for the result to be optimized through the use of a Taylor expansion, following which the partial derivative of the upper bound is solved for. The final stage of this process is the approximation obtained through inequality scaling. These methods necessitate the continual approximation of the optimal solution, and as the dimension of the channel matrix increases, the computational complexity of the solution also rises. The block coordinate descent  (BCD)method proposed in this paper attains the optimal solution by solving a single-function extremum problem. With regard to system performance indicators, the BCD scheme can circumvent the considerable computational burden of the SCA scheme, and can also obtain an acceptable confidentiality rate for user communication. It represents a compromise between complexity and reliability.

  1. How does the proposed optimization problem guarantee fairness among legitimate users in terms of their achievable secrecy rates? Maximizing the minimum achievable secrecy rate might not necessarily ensure fairness among all users. Some users might experience significantly lower secrecy rates compared to others, leading to potential dissatisfaction and security vulnerabilities.

Response: We appreciate the thoughtful review and constructive feedback provided by the reviewers. From an examination of the optimization problem (P1), it can be discerned that the objective is to achieve the maximum possible minimum confidential rate for the user. In other words, despite the presence of both interference and eavesdropping, the user is able to attain a guaranteed communication rate, with information leakage remaining within an acceptable range. All users within the system are able to fulfil their own communication and confidentiality requirements, and at this juncture, the fairness of the resources obtained by users within the system can be disregarded. Indeed, when the communication quality is within an acceptable range, no particular attention is paid to the communication rate of other users. The IRS is currently engaged in efforts to enhance wireless communication and physical layer security. However, research on the fairness of resources obtained by users in IRS-assisted communication scenarios has yet to emerge, and this represents a promising avenue for future investigation. In particular, the problem of IRS-assisted secure transmission in the context of unequal resource allocation merits further attention.

  1. How does the proposed BCD algorithm with function transformation compare to other existing methods for solving non-convex optimization problems in cell-free massive MIMO systems, particularly in terms of convergence speed, accuracy, and computational complexity?

Response: We appreciate the thoughtful review and constructive feedback provided by the reviewers. In cell-free massive MIMO, the most commonly employed convex optimization techniques include the penalty method of fractional programming (including equivalent quadratic transformation and Lagrange duality transformation), and global optimization algorithms (including branch-and-bound and branch-and-cut methods). In comparison to the aforementioned optimization techniques, the block coordinate descent algorithm proposed in this paper circumvents the necessity for successive approximation, employs a theorem to transform the problem into a single-function extremum problem, reduces the computational complexity in terms of the number of operations, and exhibits a faster convergence rate. However, in terms of calculation accuracy, it is inferior to successive convex approximation and penalty-based methods.

  1. Can you elaborate on the derivation of equation (13)? Specifically, how is the logarithm subtraction transformed into a single function maximization problem using Lemma 1?

Response: We appreciate the thoughtful review and constructive feedback provided by the reviewers. A comprehensive derivation of your question was conducted during the original process; however, due to time and space limitations, the derivation was not included in the text. To gain a deeper understanding of the specific derivation, it is recommended to consult the detailed literature sources [43] and [44].

[43] X. Guan, Q. Wu and R. Zhang, Intelligent Reflecting Surface Assisted Secrecy Communication: Is Artificial Noise Helpful or Not?, IEEE Wireless Communications Letters, vol. 9, no. 6, pp. 778-782, June 2020.

[44] Q. Li, M. Hong, H. T. Wai, Y. F. Liu, W. K. Ma, et al, Transmit Solutions for MIMO Wiretap Channels using Alternating Optimization, IEEE Journal on Selected Areas in Communications, vol. 31, no. 9, pp. 1714-1727, September 2013.

Round 2

Reviewer 1 Report

Comments and Suggestions for Authors

Accept in present form

Reviewer 3 Report

Comments and Suggestions for Authors

There is no further comments